# Effect of Wettability and Adhesion Property of Solid Margins on Water Drainage

**DOI:** 10.3390/biomimetics8010060

**Published:** 2023-02-01

**Authors:** Can Gao, Lei Jiang, Zhichao Dong

**Affiliations:** 1CAS Key Laboratory of Bio-Inspired Materials and Interfacial Science, Technical Institute of Physics and Chemistry, Chinese Academy of Sciences, Beijing 100190, China; 2School of Future Technology, University of Chinese Academy of Sciences, Beijing 100049, China

**Keywords:** solid margin, high adhesion, water channel, fast water drainage

## Abstract

Liquid flows at the solid surface and drains at the margin under gravity are ubiquitous in our daily lives. Previous research mainly focuses on the effect of substantial margin’s wettability on liquid pinning and has proved that hydrophobicity inhibits liquids from overflowing margins while hydrophilicity plays the opposite role. However, the effect of solid margins’ adhesion properties and their synergy with wettability on the overflowing behavior of water and resultant drainage behaviors are rarely studied, especially for large-volume water accumulation on the solid surface. Here, we report the solid surfaces with high-adhesion hydrophilic margin and hydrophobic margin stably pin the air-water-solid triple contact lines at the solid bottom and solid margin, respectively, and then drain water faster through stable water channels termed water channel-based drainage over a wide range of water flow rates. The hydrophilic margin promotes the overflowing of water from top to bottom. It constructs a stable “top + margin + bottom” water channel, and a high-adhesion hydrophobic margin inhibits the overflowing from margin to bottom and constructs a stable “top + margin” water channel. The constructed water channels essentially decrease marginal capillary resistances, guide top water onto the bottom or margin, and assist in draining water faster, under which gravity readily overcomes the surface tension resistance. Consequently, the water channel-based drainage mode achieves 5–8 times faster drainage behavior than the no-water channel drainage mode. The theoretical force analysis also predicts the experimental drainage volumes for different drainage modes. Overall, this article reveals marginal adhesion and wettability-dependent drainage modes and provides motivations for drainage plane design and relevant dynamic liquid-solid interaction for various applications.

## 1. Introduction

Fast water drainage, ranging from tiny droplets [1,2,3,4] to large-volume liquid [5,6,7,8], from solid surfaces is ubiquitous and critical to self-cleaning [9], water harvesting [1,2,3,4,10], and creature survival [11,12,13]. Natural surfaces, such as mosquito compound eyes [14], water strider legs [15], and drain fly tentacles [16], utilize complicated microstructures to quickly drain tiny water droplets based on surface energy or Laplace pressure gradient. Artificial surfaces can achieve preferable spontaneous and rapid water droplet removal under asymmetrical surface tension force [17,18,19,20] or external forces [21,22,23,24,25,26,27,28]. Plants’ leaves have evolved special structures, such as the drip tip apex structure [12], to rapidly remove large-volume rainwater under the gravity effect. In practice, large-volume water drainage behavior based on the overflowing of water around the solid margin can also be manipulated depending on the water flow rate and solid margin wettability [5,6,7,8,29] in a controllable manner. However, previous research mainly focuses on tiny droplet drainage at sharp spine-shaped margins or liquid columns/sheets at circle-shaped margins. Studies on the effect of the wettability and adhesion property of the macroscopic square-shaped margins on water drainage are rare, mainly when large-volume water accumulates at margins with a low slope.

Here, we establish a water channel-based faster drainage mode based on the high-adhesion hydrophilic and hydrophobic solid margins over a wide range of water flow rates. We demonstrate that the low-adhesion margins, the superhydrophobic one, the original one, and the hydrophobic one result in more significant marginal resistances and subsequent higher drainage time and drainage volumes. In contrast, the high-adhesion margins, the hydrophilic and the hydrophobic ones, construct stable water channels, quickly transport top water onto the bottom or margin, and drain water faster with less drainage time and drainage volumes. In this condition, water droplet gravity readily overcomes surface tension resistance based on the as-formed water channels. Notably, the high-adhesion hydrophobic margin sample with the highest contact angle hysteresis exhibits the fastest drainage behavior. We emphasize that stable water channels always exist for the high-adhesion margins regardless of different water flow rates, initial incline angles, and sample thicknesses. We also show the determination of the critical water flow rates versus sample thicknesses for water channel constructions associated with the original margin sample. That is, only more considerable inertia can sustain stable water channels. Further, this investigation will offer us an innovative insight into how to design structured margin planes with high-efficiency drainage behavior, especially in large-volume water accumulation situations.

## 2. Materials and Methods

### 2.1. Experimental Setup

To present the process of the water channel-based fast drainage mode, we built a drainage system including a cuboid sample connecting to a force sensor and a water injection device controlled by an injection pump (Figure 1a). First, a nozzle injects water that accumulates on the sample top. Then, the accumulated water spread on the top, and the air-water-solid triple contact line was pinned at the margin-top of the sample. At some critical conditions, the gravity component overcame the marginal surface tension resistance, and a certain volume of water was separated from the top. That is, one drainage happened. Next, constant water injection resulted in cyclical drainage behavior. Figure 1b,c shows the critical drainage images from the side view and the front view for the original margin sample, respectively. The injection needle tip was fixed at a preliminary vertical and horizontal length of *L*_v_ = 0.6 mm and *L*_h_ = 20.0 mm away from the sample surface and sample front margin, respectively, to promote the experimental stability during the water injection and drainage process. The samples were fixed at an initial incline angle of *α* = 5° and connected to the force sensor. The in-situ force sensor recorded the vertical force variations during the water injection process. An injection pump was applied to control the water flow rate *Q* ∈ {0.5–40} mL/min.

A digital camera recorded the experimental drainage process at 60 fps (Nikon D750, Japan) from the side view and a high-speed camera at 500 fps (SK 1910, Shenzhen Obeero Technology Co., Ltd., Shenzhen, China) from the front view. The in-situ vertical force *F*_z_ was measured by the universal material tester (Tribometer UMT-Tribolab, CETR, Bruker, Germany) connecting to an FL force sensor to ensure the test precision at a 50 μN force accuracy. The water flow rate *Q* ∈ {0.5–40} mL/min was regulated by an injection pump (Leadfluid TYD 02–02 Pump, China). All the water used was deionized (18.2 MΩ∙cm) from Milli-Q equipment.

### 2.2. Fabrications and Characterizations of Experimental Samples

The experimental samples were fabricated based on the polymerization reaction using the monomer methyl methacrylate via a commercial digital light processing printer (DLP) (Figure 1d) at a Z-axis resolution of 50 μm. After 3D printing, the prepared samples were first immersed in ethanol for 5 min to remove the uncured resin, dried with N_2_ gas, and then post-cured using the dry-curing equipment. Scanning electron microscope (SEM) images of the sample’s top, bottom, and front margins exhibit relatively smooth surface morphologies (Figure 1e, Appendix A). The corresponding water contact angles *θ*_top_ = 79.5 ± 2.6°, *θ*_bottom_ = 67.1 ± 1.4° and *θ*_margin_ = 78.4 ± 2.3°, respectively, showing hydrophilic features (Figure 1e, inset) of the top, bottom, and front margin of the sample. We did different margin modifications to differentiate the front margin’s wettability and adhesion properties (Figure 1f, Appendix A) and studied their effects on the water overflowing and drainage behavior. Adhesive tapes covered the top and bottom surfaces before processing different margin modifications. To make the margin more hydrophilic, the front margin was processed with O_2_ plasma (DT-03, Suzhou OPS Plasma Technology Co., Ltd., Suzhou, China) at 50 W for 5 min. To make the low-adhesion hydrophobic margin, the front margin was first processed by O_2_ plasma at 200 W for 10 min, and then the sample was put into a vacuum dryer oven at 80 ℃ for 1 h with 10 μL 1H,1H,2H,2H-Perfluorodecyltrimethoxysilane. To make the high-adhesion hydrophobic margin, the low-adhesion hydrophobic front margin was then rubbed with a sandpaper of 200 mesh for a higher contact angle hysteresis. To make the margin superhydrophobic, the front margin was carefully brushed with the superhydrophobic solution by a hairbrush and dried under ambient conditions for 5 min. We prepared the superhydrophobic solution with a mix of 0.5 g Captone ST-200 (Dupont, Shenzhen, China), 1.0 g hydrophobic fumed silica nanoparticles (Evonik Degussa Co., Frankfurt, Germany), and 30 mL ethanol. We then stirred them in a bottle for 1 h [30]. Static water contact angles and contact angle hysteresis versus different front margin modifications are summarized in Figure 1f. Except when specifically stated, the sample has a length *L* of 4.0 cm, a width *W* of 2.2 cm, and a thickness *t*_s_ of 5.0 mm.

A field-emission SEM obtained SEM images of samples at 10 kV (SU8010, Hitachi, Japan). Before SEM imaging, the samples were cleaned with water, dried with N_2_, and then sputtered with a thin layer of platinum (EM ACE, Leica, Germany). We obtained the water contact angles and water contact angle hysteresis of different margins at room temperature (LSA 100 Surface Analyzer, LAUDA Scientific, Germany). The contact angle hysteresis was measured by the sessile droplet method. A sessile drop was slowly inflated or deflated on the samples. We used a dynamic contact angle machine (DCAT 21, Data Physics, Germany) to measure the adhesion forces of different margins with 3.0 µL water droplets. Each reported data was an average of at least five independent measurements.

## 3. Results and Discussion

### 3.1. Drainage Results and Drainage Mode Comparisons

In-situ vertical force *F*_z_ results illustrate the cyclical drainage process at *Q* = 5.0 mL/min (Figure 2a). For samples with different front margin modifications, the drainage time and weight distinguish each other (Figure 2a, Appendix A). The drainage situations can be divided into no-water channel drainage (NWCD) mode and water channel-based drainage (WCD) mode. For the no-water channel drainage mode, no stable water channels exist. The drainage condition can be further divided into three circumstances according to the drainage separation zones: “T-drainage” for superhydrophobic margin-sample that inhibits water overflowing from the top to front margin and drains at the marginal top (Figure 2b, bottom left), “T+F-drainage” for low-adhesion hydrophobic margin-sample that allows water to overflow from top to front margin and then drains at the front margin (Figure 2b, bottom right), and “T+F+B-drainage” for original margin-sample that enables two successive overflowing from the top to front margin then bottom and drains at the bottom (Figure 2b, top), respectively. T, F, and B denote the sample’s top, front margin, and bottom, respectively. Selected snapshots and corresponding dashed outlines vividly demonstrate the critical drainage conditions (Figure 2b).

Water at the sample top needs to accumulate to a critical volume to overcome the barrier that emerged from the square low-adhesion superhydrophobic and hydrophobic margins. As a result, larger drainage weights collected by longer drainage time are achieved for each drainage (Figure 2a, yellow line, and purple line). For the original margin sample, a temporary, marginal water channel exists during one drainage, but it quickly dewets after the drainage. The lower-adhesion original margin fails to sustain a stable water channel during the drainage process (Appendix A). And particular, the water droplet residue at the bottom after one drainage pulls more water to drain, and in turn, leads to longer drainage time and a more extensive drainage weight (Figure 2a, black line) than that of the superhydrophobic margin sample (Figure 2a, yellow line) and low-adhesion hydrophobic margin sample (Figure 2a, purple line). During drainage, the temporary water channel at the front margin slows down the drainage process. Overall, samples with these margins drain water without stable water channels with more significant marginal surface tension resistances and larger water drainage weights.

We then focus on the water channel-based drainage mode. In this situation, stable water channels exist, and the drainage condition can be further divided into two circumstances depending on the drainage zones: “T+F+B-drainage” for the hydrophilic margin sample (Figure 2c, left) and “T+F-drainage” for the high-adhesion hydrophobic margin sample (Figure 2c, right). The hydrophilic margin permits water to overflow from top to margin and then to bottom, and the high-adhesion property facilitates stable marginal water channel construction. Then the sample drains water faster with lower drainage time and drainage weight (Figure 2a, blue line). Notably, the hydrophobic margin sample with the highest contact angle hysteresis stably captures water at the margin. It resists advancing movements to the bottom and receding movements to the top [31]. Then the marginal stable water channel assists in achieving the fastest drainage behavior (Figure 2a, red line) without contacting the bottom surface. Consequently, the water channel-based drainage mode shows a lower drainage time and drainage volume, approximately 5–8 times less than that of the no-water channel drainage mode (Figure 2d,e) for water flow rates *Q* ∈ {0.5–40} mL/min, suggesting that the faster water channel-based drainage mode applies to a broader range of water flow rates. Here, we obtain values of the drainage volume *V* as *V* = *Qt* with *Q* the water flow rate and *t* the drainage time.

### 3.2. Drainage Mechanisms and Experimental Parameters Regulating Different Drainage Modes

We next elucidate the different drainage mechanisms for the no-water and water channel-based drainage modes through detailed force analysis (Figure 3a–c). For the no-water channel drainage mode, we choose the “T-drainage” condition of the superhydrophobic margin sample as an example (Figure 3a). Water gradually accumulates on the sample top until the critical volume is achieved for water to drain. During the drainage process, the gravity component plays as the driving force *F*_d_, which scales as *ρ*V_SHB_gsinα. A marginal capillary force acts as the resisting force *F*_r,_ which scales as *γw*. At the critical drainage state, *F*_d_ equals *F*_r,_ and we obtain the drainage volume *V*_SHB_ = $\frac{{\mathrm{wl}}_{c}^{2}}{\sin\alpha }$. Through further detailed calculation, we obtain *V*_SHB_ ≈ 480 μL at *α* = 5.0°, consistent with the experimental results (Figure 2e, open yellow diamond symbol). Here, *ρ* and *γ* are the water density and surface tension, *g* is the gravitational acceleration, *l*_c_ is the capillary length, *α* is the initial incline angle, and *w* ≈ 5.7 mm is the drainage width obtained from the front-view picture (Figure 3a, inset).

In addition, we analyze the water channel-based drainage mode (Figure 3b,c). First, we consider the “T+F+B-drainage” of the hydrophilic margin sample (Figure 3b). At this condition, the water channel tightly adheres to the high-adhesion hydrophilic margin, and the injection water is quickly transferred from top to bottom via the marginal water channel. When water at the bottom surface accumulates to a critical volume, the droplet gravity *ρV*_HL_*g* exceeds the surface tension force π*γR*_j_, and one droplet drainage happens. The drainage volume is *V*_HL_ = ${\pi l}_{c}^{2}R_{j}$ and we find the *V*_HL_ ≈ 100 μL. The *R*_j_ ≈ 4.5 mm is the water jet width at the bottom surface (Figure 3b, inset). The theoretical result is slightly larger than the experimental result (Figure 2e, filled blue diamond symbol). This overestimation is rational because a slight droplet residue exists at the hydrophilic bottom surface after one drainage. Further, we consider the “T+F-drainage” of the high-adhesion hydrophobic margin one (Figure 3c). At this condition, the water channel stably pins at the high-adhesion margin without contacting the bottom surface and assists in draining via the marginal water channel. When water at the margin accumulates to some critical volume, the droplet gravity *ρV*_HA-H_*g* exceeds the surface tension force *γw_HA-H_*, and one droplet drainage happens. The drainage volume *V*_HA-H_ = $w_{\mathrm{HA}-H}l_{c}^{2}$ and we find the *V*_HA-H_ ≈ 30 μL, consistent with the experimental result (Figure 2e, filled red diamond symbol). Here, *w_HA-H_* ≈ 3.6 mm is the drainage width obtained from the front-view picture (Figure 3c, inset). Overall, stable water channels for the water channel-based drainage mode provide more convenient drainage pathways with lower marginal capillary resistances.

We investigate how the initial incline angles α affect the drainage results of both no-water and water channel-based drainage modes. We change the incline angles as *α =* 5°, 7° and 9° in a small range to guarantee a similar, more considerable water accumulation and drainage process. We confirm the power-law dependence $t\;\propto \;\frac{1}{Q\;\sin\alpha }$ under various water flow rates *Q* for samples with different margin modifications (Figure 3d). For the no-water channel drainage mode, experimental data collapse onto the master curve (Figure 3d, open diamond symbols), showing the drainage time *t* closely relates with the incline angle *α*. On the contrary, for the water channel-based drainage mode, time *t* is similar at fixed water flow rate *Q* for varying incline angle *α* under hydrophilic “T+F+B drainage” condition (Figure 3d, filled blue diamond symbols) and high-adhesion hydrophobic “T+F drainage” condition (Figure 3d, filled red diamond symbols). These results show that the incline angle α barely affects the drainage time *t* for the water channel-based drainage mode, further verifying that the stable water channel construction is the central aspect for faster drainage.

Sample thickness *t*_s_ is another crucial variable that affects drainage results under different drainage modes (Figure 3e). We define a characteristic drainage droplet size as *t*_c_ = $\sqrt[{\;3}]{\frac{{3V}_{\mathrm{SHB}}}{4\pi }}$ ~ 5.0 mm. We design the sample thickness *t*_s_ as 2.5, 5.0, and 10.0 mm with *t*_s_ = 2.5 mm smaller than *t*_c_, *t*_s_ = 5.0 mm comparable to *t*_c,_ and *t*_s_ = 10.0 mm bigger than *t*_c_. Samples with various margin modifications and drainage modes exhibit different dependences on the sample thicknesses. First, for the “T drainage” and “T+F drainage” of the no-water channel drainage mode and “T+F+B drainage” of the water channel-based drainage mode, the drainage time *t* nearly overlaps each other for various water flow rates *Q* at different sample thicknesses *t*_s_. The reasons are as follows: for the no-water channel “T drainage” and “T+F drainage” conditions, sample margins are not involved in the drainage process; for the water channel-based “T+F+B drainage” condition, stable marginal water channels construct for different sample thicknesses *t*_s_ and bottom water jet width determines the drainage results (Figure 3b). Next, for the original margin “T+F+B drainage” of the no-water channel drainage mode, threshold water flow rate *Q*^*^ exists for water channel construction and increases with *t*_s_: *Q*^*^ = 5, 10, and 20 mL/min for *t*_s_ = 2.5, 5.0, and 10.0 mm, respectively (Figure 3e, as the arrows indicate). Moreover, for the high-adhesion hydrophobic “T+F drainage” of the water channel-based drainage mode, the drainage time *t* of samples with *t*_s_ = 5.0 mm and 10.0 mm are similar, while that of the sample with *t*_s_ = 2.5 mm is larger. This is because the fully constructed water channels along thicker margins (*t*_s_ = 5.0 mm and 10.0 mm) provide more unhindered pathways to drain water, while the water channel at the thinner margin (*t*_s_ = 2.5 mm) is not fully constructed and cannot drain the injection water in time (Appendix A). These results indicate that the sample thickness largely affects the drainage results for different drainage modes.

### 3.3. Critical Drainage Snapshots and Phase Maps of the Two Drainage Modes

Finally, we analyze the drainage conditions from critical front snapshots (Figure 4a–e, Appendix A) and summarize the phase map as no-water channel drainage mode and water channel-based drainage mode versus various water flow rates *Q* for different margin modifications (Figure 4f). Critical drainage conditions vary with margin modifications. First, for the superhydrophobic margin sample with varying thicknesses, water is quickly cut off at the marginal top (Figure 4a), and no-water channel constructs because inertia (even for *Q* = 40 mL/min) cannot overcome the marginal capillary resistance (Figure 4f, open yellow diamond symbols). Then, water is also easily cut off at the margin top for the low-adhesion hydrophobic margin sample with varying thicknesses. The remaining water at the margin quickly dewets after one drainage (Figure 4b), and no marginal water exists before the next drainage for *Q* ∈ {0.5–40} mL/min (Figure 4f, open purple diamond symbols). Next, threshold water flow rates *Q** for water channel constructions exist for the original margin sample with varying thicknesses. At *Q* < *Q**, the capillary forces dominate, and the water channels gradually dewet because there is no sufficient water supply to sustain the marginal water channel (Figure 4c). At *Q* > *Q**, inertia forces dominate, and the injection water provides enough water to construct marginal water channels. Thicker margin samples need more water for water channel constructions, leading to higher *Q*^*^ (Figure 4f, open and filled black diamond symbols). Finally, water channels always tightly adhere to the higher-adhesion margins for the high-adhesion hydrophilic and hydrophobic margin samples with varying thicknesses (Figure 4d,e). Note that the drainage width for the high-adhesion hydrophilic margin is wider than the hydrophobic one (Figure 4d,e). This is because the water channel at the hydrophilic margin spreads out while the water channel at the hydrophobic one is more confined. And the samples drain water with the water channel-based drainage mode for various water flow rates *Q* ∈ {0.5–40} mL/min (Figure 4f, filled blue and red diamond symbols). In a broad aspect, lower-surface tension liquid in a realistic environment related to drainage technology can readily overflow from top to margin and then bottom [5,7] and further be drained faster based on the water channel construction. Considering the widespread distribution of exterior planes composed of intrinsically high-adhesion materials, such as glass, ceramics, metal, etc., the unveiled water channel-based faster drainage discovery in this work will undoubtedly manifest its available meanings in drainage plane design in our daily life.

## 4. Conclusions

In summary, we have presented a simple water channel-based drainage method for faster drainage based on the marginal water channel construction. The quicker water channel-based drainage mode utilizes the water channel to markedly reduce the more considerable marginal capillary resistance for the no-water channel drainage mode and exploits the gravity to overcome surface tension force readily. We confirm the effects of initial incline angles and sample thicknesses on the drainage results for samples with different margin modifications under various water flow rates for the two drainage modes. We also highlight that the sample with the highest contact angle hysteresis hydrophobic margin, previously mainly interpreted as the barrier for water transportation, exhibits the fastest drainage behavior. We believe these results provide inspiration for the widespread drainage plane design with fast water drainage behavior and may also offer insights for open high-speed water transport applications.

## Figures and Tables

**Figure 1 biomimetics-08-00060-f001:**
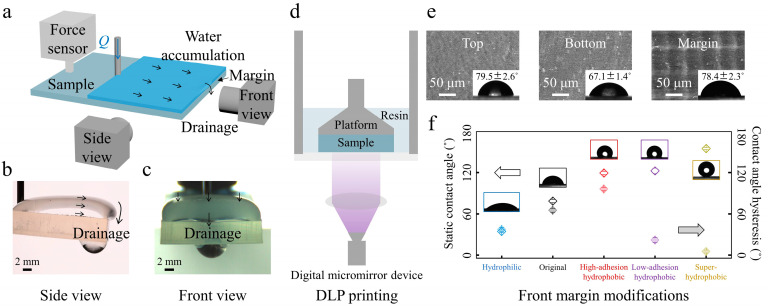
Experimental setup and material characterizations. (**a**) Schematic illustration of the experimental design. Injection water accumulates on the sample top and drains at the front margin at critical conditions. Selected essential snapshots of drainage from the side view (**b**) and the front view (**c**) for the original margin sample. (**d**) Schematic diagram of the DLP printing for sample fabrication. (**e**) Scanning electron microscope (SEM) images and corresponding water contact angles (inset) of the top surface, the bottom surface, and the front margin of the sample, respectively. (**f**) Static water contact angles and contact angle hysteresis versus different front margin modifications.

**Figure 2 biomimetics-08-00060-f002:**
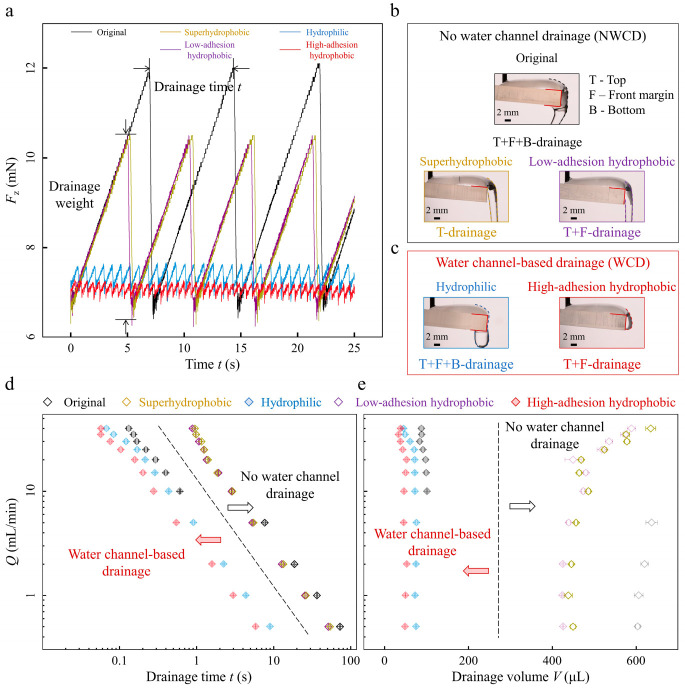
Drainage results and drainage mode comparisons. (**a**) In-situ vertical force *F*_z_ measurements versus time *t* for samples of different front margin modifications at water flow rate *Q* = 5.0 mL/min. (**b**,**c**) Critical drainage snapshots from the side view differentiate the no-water channel drainage mode (**b**) as “T+F+B drainage” ((**b**), top), “T drainage” (**b**, bottom left), and “T+F drainage” (**b**, bottom right) and water channel-based drainage mode (**c**) as “T+F+B drainage”((**c**), left) and “T+F drainage” ((**c**), right). Dashed outlines show the different critical drainage conditions. (**d**,**e**) Drainage time *t* and drainage volume *V* versus various water flow rates *Q* of the two drainage modes for samples with different front margin modifications. The error bar represents the standard deviations for at least five independent measurements.

**Figure 3 biomimetics-08-00060-f003:**
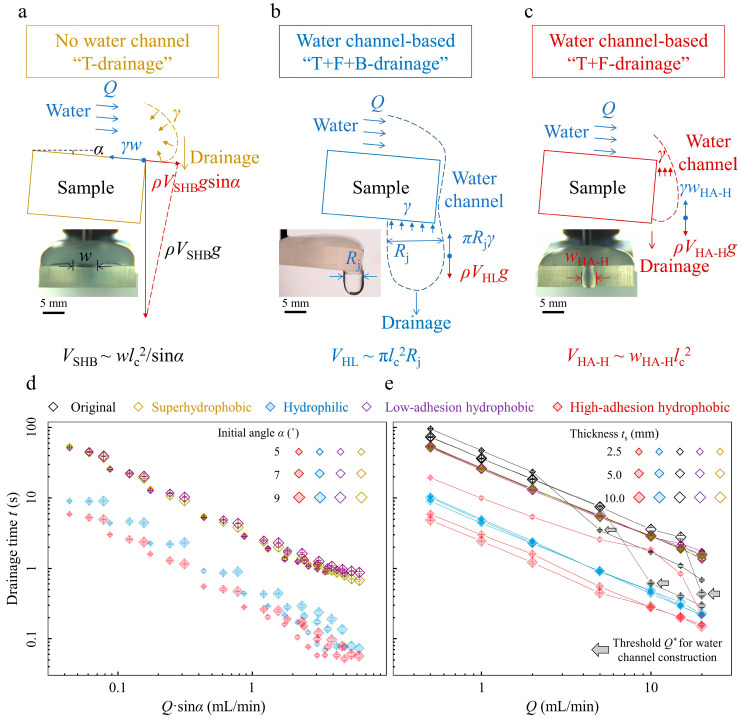
Drainage mechanisms and experimental parameters regulating different drainage modes. (**a**–**c**) Schematic diagrams and corresponding critical snapshots (inset) for different drainage modes. The driving force of gravity overcomes the resisting capillary resistance, and one droplet drainage happens. (**d**) Drainage time *t* versus water flow rate *Q* modified by the initial angle *α* for different margin modifications. (**e**) Drainage time *t* versus water flow rate *Q* under different margin thicknesses for various margin modifications. The arrows indicate the threshold water flow rates for the original margin samples. The error bar represents the standard deviations for at least five independent measurements.

**Figure 4 biomimetics-08-00060-f004:**
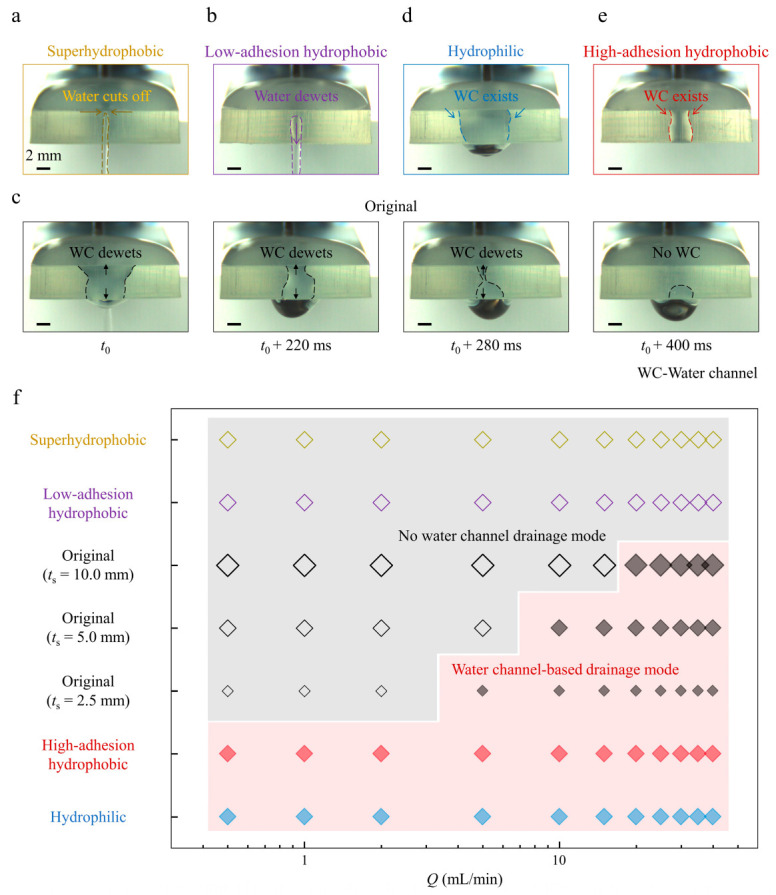
Critical drainage snapshots and phase map of the two drainage modes. (**a**–**e**) A series of critical front drainage snapshots demonstrating various marginal water channel conditions for different margin modifications. (**f**) Phase map for samples of varying margin modifications and thicknesses versus water flow rates *Q*.

## Data Availability

Not applicable.

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
