# Peer review of "Effect of Wettability and Adhesion Property of Solid Margins on Water Drainage"

_biomimetics, 2023, doi:10.3390/biomimetics8010060_

Round 1

Reviewer 1 Report

The manuscript investigates how the adhesion and wettability of solid surfaces affect water drainage. The authors systematically varied the surface properties of different sides of a solid substrate, and quantified how water flows along the solid interfaces. The work found that “water-channels” created by solid margin’s wettability and adhesion can greatly decrease the capillary resistance and improve water drainage. The paper is well written and shows interesting potential to benefit future engineering applications. I recommend the paper for publication after addressing a few minor issues listed below.

1) The label of “Drainage volume V” in Figure 2a is not appropriate since the vertical axis is the force F_z.

2) Also, related to the 1), could the authors clarify how did they measure the drainage volume from the experiments? It is not clear to me how the force (F_z) can be converted to the drainage volume.

3) From the images in Figure 4, the drainage width w is not constant for the WC-based systems? In these cases, please explain how to determine the values of w.  

Also, here are some minor issues.

4) Figure 1f is not clear. Please replace it a high-resolution image.

5) In section of Materials and Methods, could the author explain a little more on how to perform the super-hydrophobicity treatment?

Author Response

Responses to Reviewer 1

The manuscript investigates how the adhesion and wettability of solid surfaces affect water drainage. The authors systematically varied the surface properties of different sides of a solid substrate, and quantified how water flows along the solid interfaces. The work found that “water-channels” created by solid margin’s wettability and adhesion can greatly decrease the capillary resistance and improve water drainage. The paper is well written and shows interesting potential to benefit future engineering applications. I recommend the paper for publication after addressing a few minor issues listed below.

Response: Thanks very much for the reviewer’s comments. We have added more details and made thorough revisions on our manuscript. We hope that the revised manuscript would meet the reviewer’s demand and be suitable for publication in Biomimetics. Thanks again for the reviewer’s comments.

  1. The label of “Drainage volume V” in Figure 2a is not appropriate since the vertical axis is the force F_z.

Response: Thanks for the reviewer’s comments. We have removed the label “Drainage volume V” and revised the label as “Drainage weight” to be in accordance with the vertical axis Fz in Figure 2a in the revised manuscript.

  1. Also, related to the 1), could the authors clarify how did they measure the drainage volume from the experiments? It is not clear to me how the force (F_z) can be converted to the drainage volume.

Response: Thanks for the reviewer’s comments. In our experiments, we determined the drainage volume V (Figure 2e) by V = Qt with Q the water flow rate and t the drainage time. We have added relevant descriptions in Line 193-194 in the revised manuscript. The in situ measured force Fz equals ρVg and the drainage volume can also be derived as V = Fz /ρg.

  1. From the images in Figure 4, the drainage width w is not constant for the WC-based systems? In these cases, please explain how to determine the values of w.

Response: Thanks for the reviewer’s comments. For the water channel-based drainage (WCD) mode, the detailed drainage conditions are clarified as “T+F+B” drainage for the high-adhesion hydrophilic margin and “T+F” drainage for the high-adhesion hydrophobic margin. The drainage width w is not constant for the two conditions. Water channel at the hydrophilic margin spreads out while water channel at the hydrophobic margin is more confined. As a consequence, the drainage width w for the hydrophilic one is wider than the hydrophobic one (Figure 4d, e). Detailed values of w are determined by the critical drainage snapshot obtained by the front camera. We have added relevant explanations in Line 295-298 in the revised manuscript.

Also, here are some minor issues.

  1. Figure 1f is not clear. Please replace it a high-resolution image.

Response: Thanks for the reviewer’s suggestions. We have replaced the Figure 1f as a high-resolution image of 600dpi in the revised manuscript.

  1. In section of Materials and Methods, could the author explain a little more on how to perform the super-hydrophobicity treatment?

Response: Thanks for the reviewer’s suggestions. We have added relevant descriptions of superhydrophobic treatment in Line 111-113 in the revised manuscript.

Reviewer 2 Report

In this paper, the authors introduced a water channel-based faster drainage mode based on the high adhesion hydrophilic and hydrophobic margins over a wide water flow rate range. The experimental design and results are fairly good, and the paper is well presented. Therefore, this paper is acceptable in Biomimetics after the authors addressing the following questions.

 1. The part '3.1. Experimental setup and material characterizations.' should be sent to the 'Materials and methods' section and edited again.

2. Drainage technology is applied in a variety of environments. If a liquid or oil with a smaller surface tension than water, how would you expect it to behave in the system of this paper ?

3. Detailed surface information of the fabricated samples participating in the experiment is required. Authors should demonstrate additional information on surface roughness, etc. via magnified SEM images or AFM measurements.

4. minor comments : 1) Some of graphs in this manuscript are not so clearly and blur. Authors should clarify the text, number, line, and increase resolution of the graphs in all Figures for reader’s better understanding. (ex: Figure 1f) 2) Correction of typos. For example, the numbers of chapters 2 and 3 are incorrectly marked as number 1.

Author Response

Responses to Reviewer 2

In this paper, the authors introduced a water channel-based faster drainage mode based on the high adhesion hydrophilic and hydrophobic margins over a wide water flow rate range. The experimental design and results are fairly good, and the paper is well presented. Therefore, this paper is acceptable in Biomimetics after the authors addressing the following questions.

Response: Thanks very much for the reviewer’s comments. We have added more details and made thorough revisions on our manuscript. We hope that the revised manuscript would meet the reviewer’s demand and be suitable for publication in Biomimetics. Thanks again for the reviewer’s comments.

  1. The part '3.1. Experimental setup and material characterizations.' should be sent to the 'Materials and methods' section and edited again.

Response: Thanks for the reviewer’s suggestions. We have changed the place of the part '3.1. Experimental setup and material characterizations.' as the 'Materials and methods' section and edited again in the part '2.1. Experimental setup' and '2.2. Fabrications and characterizations of experimental samples.' in Line 67-127 in the revised manuscript.

  1. Drainage technology is applied in a variety of environments. If a liquid or oil with a smaller surface tension than water, how would you expect it to behave in the system of this paper?

Response: Thanks for the reviewer’s comments. Liquid with smaller surface tension (surface energy) would definitely overflow from top to (high-adhesion hydrophilic and hydrophobic) margin and then bottom1, 2 for water channel construction. As a result, the faster water channel-based drainage mode is also applicable for the lower-surface tension liquid. We have added relevant descriptions in Line 300-303 in the revised manuscript.

  1. Detailed surface information of the fabricated samples participating in the experiment is required. Authors should demonstrate additional information on surface roughness, etc. via magnified SEM images or AFM measurements.

Response: Thanks for the reviewer’s suggestions. We have added the magnified SEM images as the Figure S1 in the revised manuscript.

  1. minor comments: 1) Some of graphs in this manuscript are not so clearly and blur. Authors should clarify the text, number, line, and increase resolution of the graphs in all Figures for reader’s better understanding. (ex: Figure 1f) 2) Correction of typos. For example, the numbers of chapters 2 and 3 are incorrectly marked as number 1.

Response: 1) Thanks for the reviewer’s suggestions. We have revised the resolution of the graphs in all Figures as 600dpi in the revised manuscript. 2) Thanks for the reviewer’s suggestions. We have corrected the typos in the revised manuscript.

  1. Duez, C.; Ybert, C.; Clanet, C.; Bocquet, L. Wetting Controls Separation of Inertial Flows from Solid Surfaces. Phys. Rev. Lett. 2010, 104, 084503.
  2. Dong, Z.; Wu, L.; Wang, J.; Ma, J.; Jiang, L. Superwettability Controlled Overflow. Adv. Mater. 2015, 27, 1745-1750.

Reviewer 3 Report

The authors have carried out a careful study of the way in de-ionised water drains from an inclined flat surface under gravity examining in particular the effect of the local surface energy (and thus degree of hydrophobicity or –philicity) at or close to the plate’s edges.

Section 2.1 should include some mention of the nature of the substrate material – presumably polymeric since it was fabricated by 3D printing.

The observations indicate two flow patters NWCD (no water channel drainage) and WCD when the water flows in a well-defined channel. Is this something that happens naturally or is the flow pattern imposed in some way by modifying the margins?

Author Response

Responses to Reviewer 3

The authors have carried out a careful study of the way in de-ionised water drains from an inclined flat surface under gravity examining in particular the effect of the local surface energy (and thus degree of hydrophobicity or –philicity) at or close to the plate’s edges.

Response: Thanks very much for the reviewer’s comments. We have added more details and made thorough revisions on our manuscript. We hope that the revised manuscript would meet the reviewer’s demand and be suitable for publication in Biomimetics. Thanks again for the reviewer’s comments.

  1. Section 2.1 should include some mention of the nature of the substrate material – presumably polymeric since it was fabricated by 3D printing.

Response: Thanks for the reviewer’s suggestions. We have added some mention of the nature of the substrate material in Lines 92-94 in the revised manuscript.

  1. The observations indicate two flow patterns NWCD (no water channel drainage) and WCD when the water flows in a well-defined channel. Is this something that happens naturally or is the flow pattern imposed in some way by modifying the margins?

Response: Thanks for the reviewer’s comments. In our work, the two flow patterns of NWCD and WCD are altered by modifying the wettability and adhesion properties of solid margins. To realize the transition from NWCD to WCD, high-adhesion hydrophilic and hydrophobic margins are needed for water channel construction. In a further way, considering the widespread distribution of exterior drainage planes composed of intrinsically high-adhesion materials, such as glass, ceramics, metal, etc., the faster WCD mode can be applicable in a wide range of our daily life.

Thanks again for the suggestion and support of the reviewer. We hope that this carefully revised manuscript would meet the demands of the reviewer and be suitable for publishing in Biomimetics.